# What are the top priorities of patients and clinicians for the organization of primary cardiovascular care in Quebec? A modified e-Delphi study

Claudio Del Grande[1,2]*, Janusz Kaczorowski[1,3], Marie-Pascale Pomey[1,2]

**1** Health Innovation and Evaluation Hub, University of Montreal Hospital Research Centre, Montreal, Quebec, Canada, **2** School of Public Health, Université de Montréal, Montreal, Quebec, Canada, **3** Department of Family Medicine and Emergency Medicine, Université de Montréal, Montreal, Quebec, Canada

\* c.del.grande@umontreal.ca

**Data Availability Statement:** All relevant data are available from the Patient and clinician priorities for the organization of primary cardiovascular care in

## Abstract

### Background

Cardiovascular diseases are the leading cause of death and disability worldwide. Little is known about the organizational priorities of patients and clinicians involved in primary cardiovascular care. This study aimed to identify their shared top priorities and explore on which aspects their perspectives differed.

### Methods

A three-round modified online Delphi study was carried out with patients and clinicians in seven academic primary care settings from metropolitan, suburban and small-town areas in Quebec, Canada. Patient partners experienced in the mobilization of their experiential knowledge also participated in the study. Following an "open" round, the items elicited were assessed by a combined rating and ranking approach. Items achieving an initial consensus level ≥70% were reassessed and then rank-ordered based on their final scores. Levels of consensus achieved among patients and clinicians were compared using Fisher's Exact tests.

### Results

Thirty panelists completed the study (9 clinic patients, 7 patient partners and 14 clinicians). Out of 41 organizational aspects generated, six top priorities were shared by patients and clinicians. These related to listening and tailoring care to each patient, provision of personalized information, rapid response in the event of a problem, keeping professional training up-to-date, and relational and informational continuity of care. Statistically significant differences were found between patients' and clinicians' perspectives regarding the importance of offering healthy lifestyle and prevention activities at the clinic (lower for patients), timely access to the treating physician (higher for patients), and effective collaboration with specialist physicians (higher for patients).

Quebec [Delphi study] database (https://doi.org/10.6084/m9.figshare.20110280.v1).

**Funding:** CDG received salary support from the Dr. Sadok Besrour Chair in Family Medicine (https://medfam.umontreal.ca/recherche/chaires-fonds/la-chaire-docteur-sadok-besrour/) and received a doctoral training award from the Fonds de Recherche du Québec – Santé (https://frq.gouv.qc.ca/sante/) [award number 36266] to conduct this study. The funders had no role in study design, data collection and analysis, decision to publish, or preparation of the manuscript.

**Competing interests:** The authors have declared that no competing interests exist.

## Conclusion

Although their views differ on some organizational aspects, patients and clinicians share a small set of top priorities for primary cardiovascular care that may be transferable to other chronic diseases. These top priorities should remain a central focus of clinical settings, alongside other primary care reform goals.

## Introduction

Cardiovascular diseases and their risk factors are leading causes of morbidity and mortality globally [1, 2]. It has been estimated that approximately 90% of middle-aged and elderly individuals will develop hypertension, the leading risk factor for cardiovascular disease and premature death [3], during their remaining years of life [4]. According to the US Centers for Disease Control and Prevention, around 50% of people aged 65 years or older have prediabetes and 25% have diabetes, a condition which significantly raises the risk for cardiovascular diseases [5]. Thus, virtually everyone is likely to require cardiovascular health care in their lifetime.

In Canada, like many other jurisdictions, the routine prevention and management of cardiovascular health is carried out in primary care. Over the past decades, primary care in Canada has undergone numerous waves of reform that have led to significant changes in practice models [6]. These include: expanding group practice among family physicians; developing interdisciplinary primary care teams with allied healthcare professionals other than physicians or nurses (e.g., pharmacists, psychologists, social workers); fostering a patient-focused rather than disease-focused approach to care; emphasizing preventive and proactive care supported by clinical practice guidelines; promoting patient registries; and computerizing the health information infrastructure with the deployment of electronic medical records (EMR) [6, 7]. However, implementation of these organizational changes has been uneven across and within jurisdictions and often faces complex balancing issues for real-world clinical settings [7, 8]. For example, studies have shown that EMR use during consultations could negatively impact the communication between patient and physician and lead to unintended clinical consequences [9, 10], and that suboptimal teamwork could jeopardize informational continuity, efficiency and safety of care [11, 12].

Furthermore, it remains unclear how primary care reform goals align with the priorities of patients and clinicians, having often been decreed in "top-down" fashion by centralized health authorities [13]. Compared to the plethora of studies assessing patients' knowledge, attitudes and experiences regarding chronic illness care at the individual level, little attention has been paid to identifying their priorities at the organizational level. Over twenty years ago, Wensing et al. [14] conducted a systematic review of 57 studies on patient priorities regarding primary care. The reviewers found that most studies focused on just a few predetermined aspects of care—most often informativeness, humaneness and competence/accuracy. They reported that the aspects of care included generally came from unknown sources or were developed by the researchers themselves, casting doubt on whether they covered the full range of patients' concerns. Only a few reviewed studies (6/57) had used open-ended questions, and these did not directly assess the importance of emerging aspects other than by counting their frequencies. This is not ideal, because aspects most often thought of may not necessarily be the most important. More recently, Tran et al. solicited an e-cohort of patients in generating [15] and prioritizing [16] ideas for improving chronic illness care in France. Their approach encompassed all

chronic diseases and care settings (primary, hospital, home, community, etc.). Their initial results identified 3613 ideas related to 147 areas for improvement [15]. This is arguably an unmanageable number of issues for any clinical setting. In a following study, these results were reduced to eighteen priority areas for improvement [16], with some more relevant to primary cardiovascular care than others. However, identifying what needs to be improved is conceptually different from identifying what is essential: the former will ignore core features deemed to be sufficiently well achieved, while the latter is much less context-dependent.

During this study, the COVID-19 pandemic started and disrupted the organization of primary care, as most sectors of society. Clinical settings rapidly shifted to virtual care when routine consultations and follow-ups were postponed or cancelled due to inadequate protective resources and sudden changes in care-seeking behaviors [17–19]. There is evidence that blood pressure, glucose, and cholesterol assessments—which were less amenable to virtual visits— may have been neglected during this transition, compromising the diagnosis, treatment and follow-up of many patients [20]. As clinical settings are in the process of finding their new normal and are likely facing a backlog of cardiovascular health cases to manage, identifying the organizational priorities on which they should primarily focus to effectively meet patient needs is timely and critical. To our knowledge, no study has yet looked at the organizational priorities shared by patients and clinicians for primary cardiovascular care. Given the complexity and workload involved in primary care delivery, we believe that organizational arrangements desired by patients are unlikely to be fully implemented in clinical settings unless they are simultaneously recognized as important by the clinicians responsible for upholding them. The main objective of this study was therefore to identify the top priorities shared by primary care patients and clinicians in Quebec (Canada) to optimally organize primary cardiovascular care. Our secondary objective was to explore on which aspects of care patients' and clinicians' perspectives differed the most.

## Methods

We used a modified e-Delphi approach to conduct this study. This manuscript is written in accordance with the CREDES guideline for the reporting of Delphi studies [21].

The Delphi method is a structured approach to group communication process that aims to obtain an improved opinion from a group of knowledgeable individuals and identify areas of consensus or dissent [22]. The method is based on four main components—anonymity, iteration, controlled and systematic feedback to participants, and statistical aggregation of group response—that are intended to eliminate or reduce some distortions inherent to face-to-face discussions which can lead to biased group opinion [23]. These distortions often result from: 1) the unequal weight or influence given to participants' contributions based on their personal characteristics (e.g., looks, personality, tone of voice, reputation, seniority status) and the social context of their encounter (e.g., hierarchical relations, social desirability, face-saving); 2) the unequal amount of time allocated to discuss potential ideas, in part due to the exclusive nature of speech and path dependence of live discussion (only one person can speak at a time and preceding comments tend to shape subsequent ones); and 3) the subjective combination of opinions throughout the communication process, which remains susceptible to general impressions and oversights, personal preferences, or peer expectations. To avoid these problems, the Delphi method favors indirect communication usually performed through rounds of questionnaires. This has the added advantage of allowing participants who are geographically distant or cannot be available simultaneously to take part in the communication process.

Given the delicate professional relationships between primary care clinicians themselves (doctors, nurses, etc.) and with their patients, as well as their potential differences in social

standings, the Delphi approach offered an opportunity to bring them together in a rigorous, accessible and relatively neutral format to elicit shared organizational priorities. The asynchronous process also reduced participation requirements for busy clinicians and patients, and enabled the involvement of individuals who might otherwise have been uncomfortable in face-to-face discussions and debates with members of the other group.

## Study setting and participant selection

The study took place in academic primary care settings in Quebec, Canada. Quebec is the largest Canadian province in terms of land area and the second-largest by population, with just over 8.6 million people in 2022. Different primary care practice models coexist in the province, from traditional solo clinics to community health centers, but family medicine groups (FMG) have been the foundation for the organization of primary care since 2002 [24]. FMGs are composed of a group of family physicians (usually 6 to 12 full-time equivalent) working together and in close collaboration with registered nurses and allied healthcare professionals (e.g., psychologists, social workers, pharmacists) to care for their enrolled patients. This practice model is supported by a government funding and professional support program that is regularly updated [25]. FMGs bear many similarities to the medical home model, which serves as the basis for primary care improvement in many jurisdictions [26].

Thirteen FMG clinics affiliated with the University of Montreal's practice-based research network were invited to participate in the study, and seven accepted. Of the clinics that did not participate, one declined because it was facing major administrative changes, including a site relocation; one clinic had just come out of a tedious research experience with another team and was reluctant to commit to a new project in the short term; another was in the process of appointing its new research liaison and was unable to coordinate its participation in the project in time; and three did not respond to our invitations and showed no interest when contacted by phone. Three of the participating clinics were operating in a metropolitan area, two in suburban areas, and two were located in smaller, remote towns. These primary care settings served a diverse population of all ages and socioeconomic levels. Any adult receiving or providing cardiovascular care at the participating sites was eligible to participate in the Delphi process. For patients, this included persons with established cardiovascular disease such as heart disease or heart failure, or diagnosed with a cardiovascular risk factor such as hypertension or dyslipidemia. For clinicians, this included family physicians, nurses, and all on-site allied healthcare professionals involved in the cardiovascular care of patients. In addition to on-site patients and clinicians, we also invited patient partners affiliated with the Patient Collaboration and Partnership branch of our university's Faculty of Medicine [27] to take part in the study. These patient partners are formally trained and experienced in the mobilization of their experiential knowledge to improve care and health research alongside professionals [28]. As such, we felt that their inclusion would enhance the quality of the communication process. To be eligible, patient partners were not required to be followed in one of the participating FMGs but had to be receiving primary cardiovascular care in a FMG setting. Because data collection took place exclusively online, interested individuals were required to have access to a computer, tablet or smartphone with an internet connection to participate in the study.

## Panel size

The Delphi approach differs from conventional quantitative surveys in that it aims to refine a group's opinion over the course of several rounds rather than to provide a "one-off" representation of it. Sharing reasons in support of one's assessments is the essential means of circulating valuable knowledge among Delphi panelists to improve group response between rounds [29].

Like ordinary surveys, Delphi panels that are too small are more likely to deliver less accurate and reliable results, particularly if varied perspectives are relevant to the topic of interest [30, 31]. However, unlike ordinary surveys, larger Delphi panels can become unwieldy and suboptimal in terms of logistical and analytical resources required, due to the high volume of material needed to be reviewed after each round [31, 32]. Maintaining a personalized approach to communicate with participants to sustain their engagement and reduce drop-outs is also more challenging with larger panels [31, 32]. Evidence suggests that between 5–20 panel members is generally adequate for Delphi studies [32]. In a systematic review of 80 Delphi studies in healthcare, the median number of panel members was found to be 17 (interquartile range = 11–31) [33]. Considering the heterogeneity of lay and professional expertise sought in our study, we aimed to recruit 40 participants, roughly balanced between patients and clinicians, to obtain a panel size of about twice that median number assuming a 15% loss to follow-up.

## Participant recruitment

Recruitment at participating clinics was carried out using posters and flyers posted on bulletin boards in waiting rooms and staff rooms. Additionally, the study was presented to clinicians during lunch conferences, and two whole-day visits were made at each site at different times of the week to distribute flyers to all adult patients in attendance. Eligible faculty patient partners were sent an invitation email with the recruitment flyer by a coordinator of Université de Montréal's Patient Collaboration and Partnership branch. Interested individuals had to confirm their eligibility and give consent online after reviewing the study's information and consent form. An incentive of entering a draw for a 10% chance of winning an Amazon gift card worth $25 CAD was offered to participants who would complete all study questionnaires.

The study was approved by the University of Montreal Hospital Research Centre's research ethics committee (project number 17.305). Participant recruitment and data collection took place over a one-year period, from November 2019 to November 2020.

## Modified e-Delphi process

Online questionnaires were administered using the SurveyMonkey platform. We decided in advance to stop the Delphi process after three rounds, including a first "open" round, which represents the minimum for participants to benefit from group feedback on item assessments. Like others [33, 34], we felt that additional rounds were unlikely to introduce significant changes in top priorities and were not worth the risk of increasing attrition rates due to the repetitive nature of the exercise. All study questionnaires were designed to be completed in less than 20 minutes each. Personalized follow-ups were provided to panel members throughout the study, with up to three reminders sent to nonrespondents two weeks apart during each round. Study materials were available in English and French. All questionnaires were pretested with nonparticipating patients (n = 2) and clinicians (n = 2 family physicians) to ensure readability and clarity (shortening and simplification of words and sentences), and to estimate completion time. Details on data collection and analysis are reported in the following paragraphs on a per-round basis. Qualitative data analyses were conducted in Word and Excel software (Microsoft Corp.), and statistical analyses were performed in SPSS version 26 (IBM Corp.).

## Round one: Item generation

The first study round was "open" to allow participants to express their views as freely as possible and avoid imposing from the onset an academic framing of the organization of care.

Participants were asked: "What are the priority aspects to ensure that cardiovascular care in primary care settings is organized in the best way possible for patients?". Summary definitions of cardiovascular care and primary care were given to ensure that panelists would consider the full scope of primary cardiovascular care (prevention, diagnosis, treatment and ongoing management) and that they would not refer to care received in hospital or other specialized care settings. Panelists were explicitly advised to think about the organizational elements they felt were essential for patients, whether these were already implemented in their own primary care setting or not. Each panelist could provide up to 5 responses in free-text fields. Questions about participants' age group, sex, area of residence, education level, main activity, general health, and chronic conditions followed afterwards.

Free-text responses were analyzed by the main author to elicit distinct items that would provide specific guidance to primary care clinical settings. Elements that were outside their scope of action were excluded (e.g., training more cardiologists, or implementing a single access point at the regional level to manage referrals to specialist physicians). These categorizations were validated by the other two authors. The items were grouped into thematic lists to facilitate their assessment during the second round. Item labels were revised and finalized with our pretest participants.

### Round two: Initial assessment of items

In round two, panelists assessed the items generated for the first time. It was felt that rating and ranking would provide valuable complementary insights to identify a concise set of priorities. Ratings give an order of magnitude as to how far apart two items are in terms of appreciation, but discriminate poorly among items of similar importance. This often results in large numbers of items retained, as seen in other Delphi studies [35–37]. Conversely, ranking forces the prioritization of items, but treats the gap between consecutively ranked items as equivalent throughout the list. However, this is rarely the case for any respondent and can lead to adopting spurious cut-off points. To benefit from the synergy of these two assessment techniques, avoid redundancy and limit the time required to complete the questionnaires, participants were randomly allocated to a rating or ranking subpanel stratified by their sex (female, male) and status (clinic patient, faculty patient partner, clinician) to maintain balance between both groups.

In the rating subpanel, participants assessed the items on a 7-point unipolar scale ranging from "1-Not at all important" to "7-Extremely important". The number of categories on the scale was based on previous research summaries indicating that reliability and validity tend to increase up to seven response options [32, 38, 39]. Verbal and numeric labels accompanied each response category to reduce measurement error (Fig 1) [32, 39]. In the ranking subpanel, participants ranked the items in descending order of importance in their thematic list. To prevent any order effects, thematic lists and items within each list were presented in random order to each panelist in both subpanels. Participants were encouraged to provide additional explanations (reasons) to support their assessments in free-text fields when they judged an item to be essential or, conversely, expandable. Providing explanations was not compulsory,

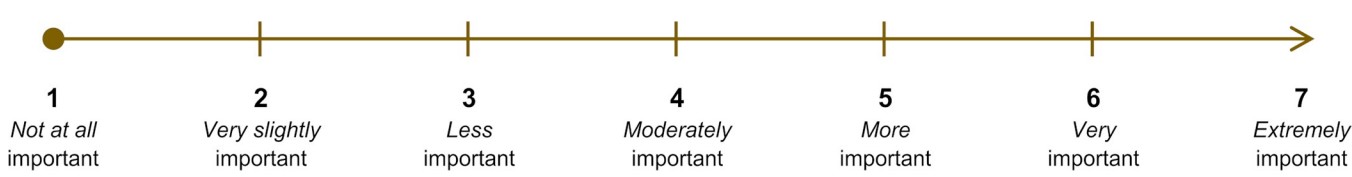

**Fig 1. Response scale for rating the items.**

but the questionnaire emphasized that it was a crucial component to allow others in their sub-panel to better consider their opinion and improve the quality of the end results.

At the end of the second round, the explanations provided were summarized and rephrased in a neutral manner so that it could not be determined whether they were formulated by patients or clinicians. They were then grouped into positive or negative comments for each item. Median ratings and rankings as well as the interquartile range (IQR) of responses were calculated. Level of consensus to be retained as a top item was determined from the overall proportion of panelists either rating an item as "6-Very important" or "7-Extremely impor-tant" in the rating subpanel or ranking the item in the top half of its thematic list in the ranking subpanel. Focusing on the highest response categories on the rating scale or top half within a ranked list is common in Delphi studies to define consensus [40, 41]. The criteria chosen for the rating and ranking subpanels were deemed sufficiently comparable because we expected that most ratings would fall between 4 and 7 on the response scale, given that all the items had been freely elicited as important by panelists beforehand. The consensus threshold for an item to be reassessed in the third round was set at 70%. While not too strict, this threshold ensured that panelists would focus exclusively during the last round on the items most susceptible of making it into the final top priorities. Consensus levels achieved among patients and clinicians for each item were compared using Fisher's Exact tests (two-tailed significance) due to small sample sizes. Statistical significance was set at $p < 0.05$.

### Round three: Final assessment of top items

In the third questionnaire, panelists were asked to provide an updated assessment of the most important organizational items after taking into account the feedback provided. This feedback included, for each item, the median rating or ranking and IQR, the structured summary of positive and negative comments made by other participants in their subpanel, and the panel-ist's initial assessment. This time, the items were presented as a single set, in descending order of importance based on their median rating or ranking, as recommended in Delphi methodol-ogy [41]. This also meant that ranking panelists could directly prioritize items that were previ-ously isolated in different thematic lists. However, ranking a relatively large number of items poses a greater cognitive burden compared to rating [42]. To mitigate this, panelists in the ranking subpanel first selected the items they considered to be in their top half from the full list after reviewing the feedback provided. They then proceeded to rank the items in their top and bottom halves separately, with the full ranking reconstructed at the analysis stage. In an optional section at the end of the questionnaire, identical feedback was presented to panelists for the items that had not reached the 70% threshold in the previous round. Statistical analyses for round three data were identical to those performed after round two. To identify the top organizational priorities, the items reassessed were rank-ordered based on their final consen-sus level.

## Results

Fig 2 depicts the study flow. Forty-one individuals gave their consent to participate. Two were found to be ineligible and three were lost to follow-up during the first round. Thus, 36 panel-ists actively took part in the study: 20 patients (13 clinic patients, 7 faculty patient partners) and 16 clinicians (10 family physicians, 5 registered nurses and 1 allied healthcare profes-sional). The median number of panelists recruited from each of the seven participating clinics was 4 (min = 2, max = 7).

Panelists' characteristics are presented in Table 1. Patients were evenly balanced between women and men. They spanned across different age groups and almost all had a post-

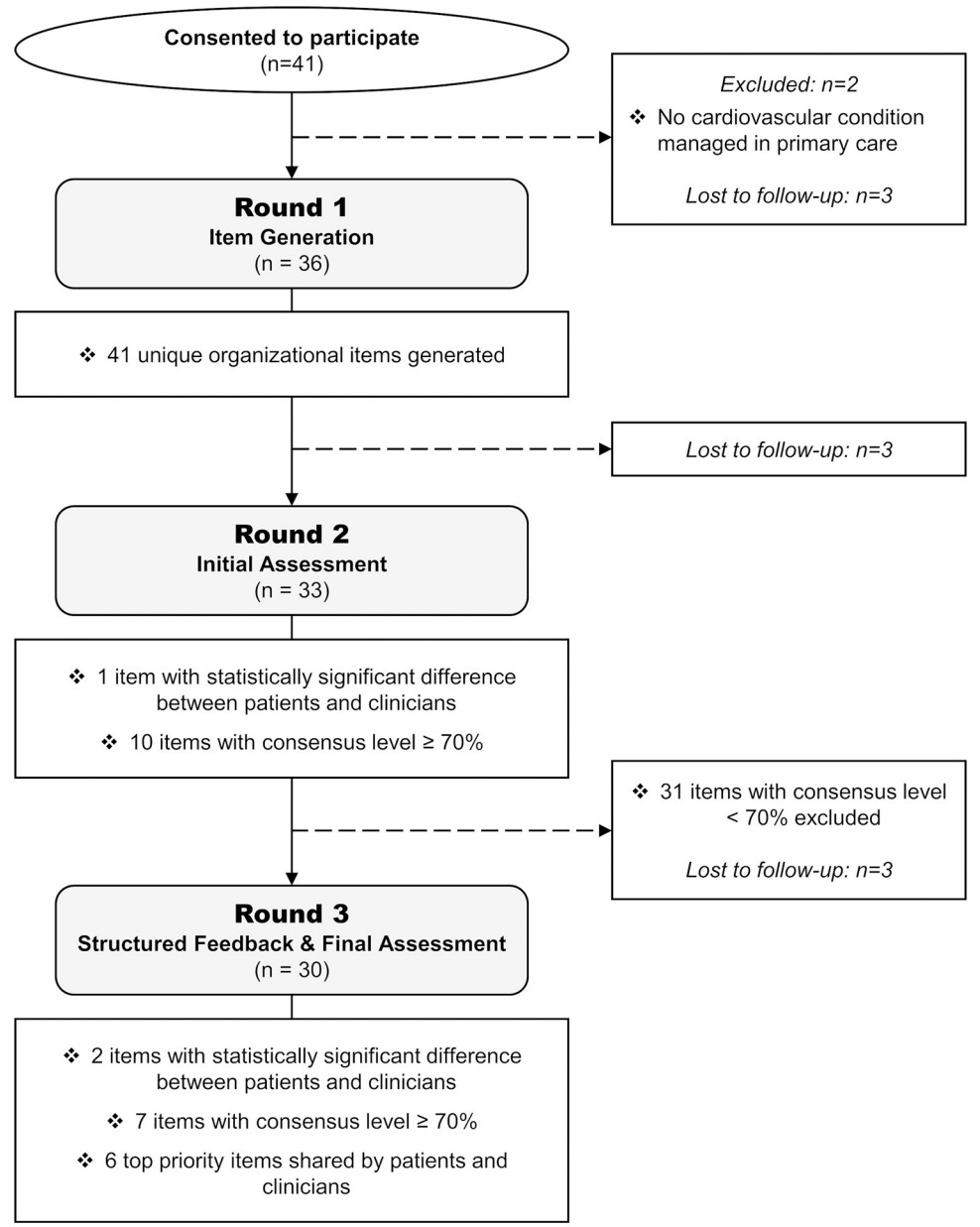

**Fig 2. Flow chart of the Delphi process.**

secondary education. The vast majority had multimorbidity—a common phenomenon in primary care [43]—and half of them had been living with a chronic condition for more than 15 years. Compared to patients, the participating clinicians were more likely to be female, young, educated and healthy. Thirty panelists (9 clinic patients, 7 faculty patient partners, and 14 clinicians) completed all three Delphi rounds (83% retention rate following completion of the first round).

## Round one

During round one, 149 potentially important organizational aspects were submitted: 78 by patients (mean = 3,9 per patient panelist; min = 1, max = 5) and 71 by clinicians (mean = 4,4

**Table 1. Characteristics of Delphi panelists.**

| | First round | | Third (final) round | |
|---|---|---|---|---|
| | Patients[a] | Clinicians | Patients[a] | Clinicians |
| | n = 20 | n = 16 | n = 16 | n = 14 |
| **Setting** | | | | |
| Metropolitan | 9 (45%) | 8 (50%) | 6 (37.5%) | 6 (42.9%) |
| Suburban | 7 (35%) | 2 (12.5%) | 7 (43.8%) | 2 (14.3%) |
| Smaller, remote town | 4 (20%) | 6 (37.5%) | 3 (18.8%) | 6 (42.9%) |
| **Gender** | | | | |
| Female | 11 (55%) | 10 (66.7%) | 10 (62.5%) | 10 (71.4%) |
| Male | 9 (45%) | 5 (33.3%) | 6 (37.5%) | 4 (28.6%) |
| **Age group** | | | | |
| 18–34 | 1 (5%) | 10 (66.7%) | 1 (6.3%) | 10 (71.4%) |
| 35–49 | 3 (15%) | 4 (26.7%) | 2 (12.5%) | 4 (28.6%) |
| 50–64 | 7 (35%) | 1 (6.7%) | 7 (43.8%) | 0 (0%) |
| 65–79 | 6 (30%) | 0 (0%) | 4 (25%) | 0 (0%) |
| 80+ | 3 (15%) | 0 (0%) | 2 (12.5%) | 0 (0%) |
| **Education level** | | | | |
| High school | 2 (10%) | 0 (0%) | 2 (12.5%) | 0 (0%) |
| College / Vocational | 9 (45%) | 0 (0%) | 6 (37.5%) | 0 (0%) |
| University | 7 (35%) | 15 (100%) | 6 (37.5%) | 14 (100%) |
| Preferred not to answer | 2 (10%) | 0 (0%) | 2 (12.5%) | 0 (0%) |
| **Perceived general health** | | | | |
| Excellent or very good | 5 (25%) | 12 (80%) | 3 (18.8%) | 11 (78.6%) |
| Good | 12 (60%) | 3 (20%) | 10 (62.5%) | 3 (21.4%) |
| Fair or poor | 3 (15%) | 0 (0%) | 3 (18.8%) | 0 (0%) |
| **Number of chronic conditions[b]** | | | | |
| 0 | 0 (0%) | 8 (53.3%) | 0 (0%) | 7 (50%) |
| 1 | 5 (25%) | 4 (26.7%) | 3 (18.8%) | 4 (28.6%) |
| 2–3 | 8 (40%) | 3 (20%) | 7 (43.8%) | 3 (21.4%) |
| 4+ | 7 (35%) | 0 (0%) | 6 (37.5%) | 0 (0%) |
| **Years of living with a chronic condition** | | | | |
| Less than 5 years | 4 (20%) | 2 (13.3%) | 2 (12.5%) | 2 (14.3%) |
| Between 5 and 15 years | 6 (30%) | 3 (20%) | 6 (37.5%) | 3 (21.4%) |
| More than 15 years | 10 (50%) | 2 (13.3%) | 8 (50%) | 2 (14.3%) |

Percentages in each cell do not account for missing values.

[a]Patients encompass both clinic patients and faculty patient partners.

[b]Measured from a list of 12 common physical and mental long-term health conditions with "Other–please specify".

per clinician panelist; min = 3, max = 5). They were synthesized into 41 mutually exclusive items and grouped under 9 themes (Table 2). Of those, twenty-seven items had been elicited by both patient and clinician panelists, 9 items by patients only, and 5 by clinicians only. The most significant change made in item definition during the pretest was to combine listening to patients and tailoring care to each patient into a single relational item (PPR1). Our pretest patients were adamant that genuine listening always reflected in the care provided, and that it was impossible to effectively tailor care to each patient without true listening. Videoconferencing as a modality for accessing a professional in the event of a problem in item A2 was also added in the context of the COVID-19 pandemic.

**Table 2. Organizational items generated during round one, grouped in themes.**

**Accessibility:** *whether it is easy and convenient to obtain services at the clinic*
- {A1} Being able to get an appointment with your family doctor on short notice.
- {A2} Being able to reach a healthcare professional within 24–48 hours in the event of a problem, either on site, by phone, videoconference or email.
- {A3} Having the option to get longer consultations.[a]
- {A4} Having access to all clinic services in the evening and on weekends.[a]
- {A5} Having access to all clinic services in French or English.[a]
- {A6} Being seen on time for an appointment with little or no delay.[b]
- {A7} Free parking near the clinic.[b]

**Services Network:** *whether the clinic is well connected with other resources in the healthcare system*
- {SN1} Obtaining short delays for examinations and consultations that must be done outside the clinic.
- {SN2} Having access to inexpensive or free resources and programs to improve the health and lifestyle of people with a cardiovascular health condition.
- {SN3} Having access to a variety of tests (blood tests, echocardiography, etc.) at the clinic without having to be referred externally.
- {SN4} Coordinating the appointments (in and out of the clinic) to minimize the inconvenience to patients.[b]
- {SN5} Explaining the role of each healthcare professional and when/how to refer to the right person.

**Care and Follow-Up:** *whether the care provided at the clinic is proactive and preventive*
- {CFU1} Having protocols in place to systematically direct patients to the right care and services based on their condition.
- {CFU2} Conducting regular follow-ups on the progress made or not (e.g., in a logbook detailing steps of care).[b]
- {CFU3} Offering activities at the clinic on healthy lifestyle and prevention of cardiovascular health problems.
- {CFU4} Offering help in managing health-related stress and anxiety.[b]

**Self-Management Support:** *whether patients are supported in understanding their health condition and caring for themselves*
- {SMS1} Receiving general information on cardiovascular health and available support resources.
- {SMS2} Receiving personalized information on your own cardiovascular health (personal check-up, origin and nature of the problem, risks, etc.).[b]
- {SMS3} Receiving training and tools to help you manage your own health (how to take your blood pressure, what to do based on your results, etc.).
- {SMS4} Receiving practical help to initiate lifestyle changes (nutritional evaluation, health literacy education service, etc.).

**Clinical Team Composition:** *whether the clinical team includes professionals from different disciplines*
- {CTC1} Having a pharmacist available on the clinical team.
- {CTC2} Having a nutrition specialist available on the clinical team.
- {CTC3} Having a physical activity specialist available on the clinical team.
- {CTC4} Having a specialist in weight and obesity management available on the clinical team.
- {CTC5} Having a smoking cessation specialist available on the clinical team.
- {CTC6} Having a nurse specialized in cardiovascular health available on the clinical team.

**Professional Collaboration:** *whether healthcare professionals work together effectively*
- {PC1} Ensuring effective collaboration between family doctors and nurses at the clinic.
- {PC2} Ensuring effective collaboration between the clinic and pharmacists in the community.
- {PC3} Ensuring effective collaboration between family doctors and allied healthcare professionals specializing in healthy lifestyles.
- {PC4} Ensuring effective collaboration between the clinic and specialist physicians (e.g., cardiologists).
- {PC5} Ensuring effective collaboration between the clinic and community resources.

**Professional Training:** *whether healthcare professionals are well trained and experienced*
- {PT1} Healthcare professionals having up-to-date cardiovascular health training in their respective fields.
- {PT2} Having a doctor who is better trained to provide counseling on nutrition and physical activity.
- {PT3} Having a doctor with sufficient clinical experience in cardiovascular health.[b]

**Patient-Professional Relationship:** *whether the relationship between patients and healthcare professionals is sincere and seamless*
- {PPR1} Feeling that healthcare professionals are truly listening in order to tailor care according to the motivation and requests of each patient.
- {PPR2} Involving the patient's family and loved ones in care.[a]
- {PPR3} Ensuring consistency in the professionals who follow the patient (same doctor, same nurse, etc.).

**Information Systems:** *whether the information systems used at the clinic are responsive and advanced*
- {IS1} Easy access for patients to their medical records.[b]
- {IS2} Having a dedicated phone support line for registered patients where nurses would have access to the patients' records.[b]
- {IS3} Being able to send and receive information electronically with the clinic (email, texting) regarding health status, test results, notifications for follow-ups, etc.
- {IS4} Having a single, common medical record between all healthcare providers.[a]

[a]Elicited by clinician panelists only.

[b]Elicited by patient panelists only.

The items most frequently mentioned by our panelists overall were "being able to reach a healthcare professional within 24–48 hours in the event of a problem, either on site, by phone, videoconference or email" (by 33.3% of panelists), "feeling that healthcare professionals are truly listening in order to tailor care according to the motivation and requests of each patient"

(by 30.6%), "having a nutrition specialist available on the clinical team" and "having a nurse specialized in cardiovascular health available on the clinical team" (by 27.8% each). Compared to patients, a higher proportion of clinicians elicited at least one item in the clinical team composition (62.5% vs. 20.0%, Fisher-exact p = 0.016) or the services network (75.0% vs. 40.0%, p = 0.049) themes.

## Round two

Thirty-three panelists (17 patients and 16 clinicians) completed round two. Two-thirds of them (21/33) shared additional explanations in free-text to support their assessments. Ten items achieved a consensus level of 70% or greater and were retained for the final round (min = 72.7%, max = 90.6%). None of the items in the "Care and follow-up" or "Clinical team composition" themes met the designated consensus threshold. The least prioritized items were: "receiving general information on cardiovascular health and available support resources" (consensus = 15.6%); "involving the patient's family and loved ones in care" (28.1%); "easy access for patients to their medical records" (31.3%); "having a smoking cessation specialist available on the clinical team" (31.3%); and "having the option to get longer consultations" (31.3%). The quantitative results for the 41 items assessed during round two can be found in S1 Table.

Only one statistically significant difference was observed between patients and clinicians during this round. The item "offering activities at the clinic on healthy lifestyle and prevention of cardiovascular health problems" was more favored by clinicians than patients (60.0% vs. 17.6%, Fisher-exact p = 0.027), but the consensus level was still below the 70% threshold in both groups of panelists. Negative comments indicated that activities on healthy lifestyle and prevention could be better accomplished outside the clinic by community centers or dedicated teams at the regional level, and that some patients could feel annoyed or embarrassed with the repeated focus put on their lifestyle habits in the clinic, especially since many may have already had mixed experiences with counseling.

## Round three

Fig 3 shows an example of the structured feedback provided in round three. The final results for the ten organizational items that had obtained an initial consensus level of at least 70% and were reassessed during the final round are presented in Table 3. The number one top organizational priority, supported unanimously by our panelists, was "feeling that healthcare professionals are truly listening in order to tailor care according to the motivation and requests of each patient". Six other items achieved a final consensus level of 70% or greater. These were, in descending order: "receiving personalized information on your own cardiovascular health" (86.7%); "being able to reach a healthcare professional within 24–48 hours in the event of a problem" (80.0%); "ensuring consistency in the professionals who follow the patient" (80.0%); "healthcare professionals having up-to-date cardiovascular health training" (76.7%); "having a single common medical record between all healthcare providers" (73.3%); and "being able to get an appointment with your family doctor on short notice" (70.0%). However, this last item was not a priority shared by patients and clinicians (consensus = 93.8% for patients vs. 42.9% for clinicians, Fisher-exact p = 0.004).

Another statistically significant difference between patients and clinicians was about "ensuring effective collaboration between the clinic and specialist physicians", which was more favored by patients than clinicians (68.8% vs. 21.4%, Fisher-exact p = 0.014). This difference was already hinted at during the previous round (88.2% initial consensus for patients vs. 56.3% for clinicians, Fisher-exact p = 0.057). Positive comments about this item revealed, on one

**ITEM:** Receiving personalized information on your own cardiovascular health
(personal check-up, origin and nature of the problem, risks, etc.)

**Distribution of initial ratings:**

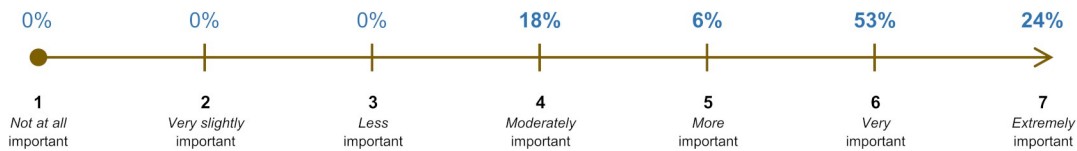

| 0% | 0% | 0% | **18%** | **6%** | **53%** | **24%** |
|---|---|---|---|---|---|---|
| **1** | **2** | **3** | **4** | **5** | **6** | **7** |
| *Not at all important* | *Very slightly important* | *Less important* | *Moderately important* | *More important* | *Very important* | *Extremely important* |

**Central tendency: 6**

**Dispersion: 1.0**

**Positive comments: (4 participants)**

- This is important to help patients get involved in their health.
- This is important to increase patient autonomy, help patients better understand how to monitor their situation and adapt, and be on the lookout for the causes of problems so they don't repeat them.
- Personalized information is valuable and even more relevant because unlike general information, it is not found on websites.

**Negative comments: (2 participants)**

- This is less important because some patients don't want to know too much about their condition, otherwise they get worried.

**Your _initial_ rating was: 6**

**\* Please select your _final_ rating: [ ]**

**Fig 3. Example of the structured feedback provided to panelists during round three.**

**Table 3. Quantitative results for the top organizational items reassessed during round three, rank ordered based on final consensus (n = 30 panelists).**

| Rank | Organizational items | Final consensus[a] | Difference[b] between [P]atients and [C]linicians | Change in consensus from previous round |
|---|---|---|---|---|
| 1 | {PPR1} Feeling that healthcare professionals are truly listening in order to tailor care according to the motivation and requests of each patient | 100.0% | N/A | +9.4% |
| 2 | {SMS2} Receiving personalized information on your own cardiovascular health (personal check-up, origin and nature of the problem, risks, etc.) | 86.7% | N.S. (p = 1.000) | +5.4% |
| 3 | {A2} Being able to reach a healthcare professional within 24–48 hours in the event of a problem, either on site, by phone, videoconference or email | 80.0% | N.S. (p = 1.000) | -7.5% |
| 3 | {PPR3} Ensuring consistency in the professionals who follow the patient (same doctor, same nurse, etc.) | 80.0% | N.S. (p = 0.657) | -10.6% |
| 5 | {PT1} Healthcare professionals having up-to-date cardiovascular health training in their respective fields | 76.7% | N.S. (p = 1.000) | -10.8% |
| 6 | {IS4} Having a single, common medical record between all healthcare providers | 73.3% | N.S. (p = 0.689) | -8.0% |
| 7 | {A1} Being able to get an appointment with your family doctor on short notice | 70.0% | **P [93.8%] > C [42.9%] (p = 0.004)** | -11.3% |
| 8 | {PC1} Ensuring effective collaboration between family doctors and nurses at the clinic | 60.0% | N.S. (p = 0.284) | -24.8% |
| 9 | {SN4} Coordinating the appointments (in and out of the clinic) to minimize the inconvenience to patients | 50.0% | N.S. (p = 0.272) | -28.1% |
| 10 | {PC4} Ensuring effective collaboration between the clinic and specialist physicians (e.g., cardiologists) | 46.7% | **P [68.8%] > C [21.4%] (p = 0.014)** | -26.0% |

[a]Based on the proportion of panelists rating the item as either 6-"very" or 7-"extremely" important (rating subpanel), or ranking the item in the top half of the list (ranking subpanel).

[b]Fisher's Exact tests, two-tailed significance.

N/A: not applicable.

N.S.: not statistically significant (p ≥ 0.05); P: patients; C: clinicians.

hand, that effective collaboration with specialist physicians was seen as important for patients to avoid having to tell their medical history repeatedly; for easing their burden of follow-up as some patients have multiple problems and are followed by many specialists; and for family and specialist physicians to coordinate their treatments and actions. On the other hand, it was also suggested that lack of communication rather than of in-depth collaboration was the main issue at hand, and that simply ensuring that information flowed easily in both directions would be adequate.

## Discussion

This study used a rating and ranking Delphi approach to identify a small set of priorities shared by patients and clinicians to organize primary cardiovascular care in the best way possible for patients in Quebec FMG clinics. Over a three-round process, thirty panelists, including regular clinic patients, experienced patient partners, and clinicians, elicited and then assessed the importance of 41 organizational aspects. These largely covered the aspects related to the organizational structure, resources and processes of care at the clinic level found in the primary care and chronic illness care literature [44, 45]. This is a testament to the collective expertise of our panel and reduces the possibility that an important organizational feature was missed.

The top priorities shared by patients and clinicians in this study were about genuine listening by clinicians to tailor care for each patient, receiving personalized information about one's health, being able to reach a healthcare professional from the clinic quickly in the event of a problem, ensuring relational and informational continuity of care, as well as keeping professional skills up-to-date. These priorities are largely consistent with previous research investigating important aspects of care for patients. In Wensing et al.'s 1998 systematic review, the most important aspects for patients regarding general practice across the 19 studies that had importance rank-orders were humaneness, competence/accuracy, patients' involvement in decisions, time for care, availability/accessibility, informativeness, and exploring patients' needs [14]. Aside from "having the option to get longer consultations" which was among the least prioritized items in our study, our top shared priorities strongly reflect these aspects of care. Although none of our panelists explicitly used the word 'decision' during our first round, the idea of healthcare providers welcoming and heeding patients' requests to inform care decisions is salient in the number one top shared priority coming out of our study. This item also encapsulates the aspects of humaneness and exploring patients' needs present in Wensing et al.'s review, underlining the interconnections between these fundamental aspects of care. Our results are also consistent with those from Tran et al.'s contemporary citizen science studies [15, 16], where creating the context for real discussions between patients and physicians, informing patients about their own care, personalizing treatment based on patient preferences, improving access to emergency care, and improving professionals' knowledge were among the most elicited and prioritized ideas from patients seeking to improve chronic illness care in France. Our findings strengthen this evidence base by reaffirming the preeminent role of these organizational aspects for patients in our own jurisdiction and in the specific context of primary cardiovascular care. They also add to this literature by showing that primary care clinicians also strongly support them.

Moreover, our study highlighted interesting contrasts between the organizational perspectives of patients and clinicians. When asked about what is important to organize care optimally for patients, our clinician panelists more readily mentioned the contribution of other allied healthcare professionals and of the broader network of health services around the clinic. This may reflect clinicians' heightened sense of need for external support to provide

comprehensive, whole-person care to their patients. Patients, on the other hand, may be more concerned about disruptions in continuity of care when new professionals are added to their care team. In a cluster randomized controlled trial to test the impact of involving patients in setting healthcare improvement priorities at the community level, Boivin et al. [46] had also found that interdisciplinary teams was a higher priority for health professionals than patients. Another difference found was that our patient panelists put less importance than clinicians on wanting their clinic to be actively involved in the promotion of healthy lifestyle habits. In Boivin et al.'s study [46], physical activity counseling was also a higher priority for professionals than patients. Furthermore, Beaulieu et al. [47] had similarly found that patients viewed monitoring weight and smoking much less favorably than professionals in the context of developing quality indicators to support chronic disease management. Lifestyle habits are a key modifiable risk factor for cardiovascular conditions, but sustainable change is hard to achieve and is influenced by numerous individual and environmental factors over which clinical settings may have little control. The explanations provided by our panelists indicate that many patients may harbor mixed feelings about focusing on their lifestyle choices in the primary care clinical setting. Our patient panelists also placed a much higher priority than clinicians on timely access to their treating physician, a difference similarly found by Boivin et al. [46] Access to family physicians is an important challenge in Canada and especially in Quebec, where about 20% of the population do not have a family physician and those that do often struggle to book an appointment on a short notice [48, 49]. We did not obtain enlightening reasons to explain this difference in our study. However, family physicians, who made up most of our clinician panelists, may have dreaded the pressure to make themselves available at short notice. The fact that our clinicians were recruited from university-affiliated clinics, where they are required to manage academic duties (teaching, supervision, research) on top of their clinical responsibilities, may have also accentuated their reluctance towards this item. In comparison, providing quick access to a professional in the event of a problem may have been seen as less of a threat because it implied team responsibility. Finally, a higher proportion of our patient panelists compared to clinicians viewed the collaboration between their clinic and specialist physicians to be a key organizational aspect. Patients may expect more from their primary care setting regarding this collaboration than what many clinicians would consider sufficient.

The results of this study should be of interest to practice facilitators, managers, policy makers, clinicians and patients involved in primary care improvement in Quebec and beyond. Firstly, knowing which aspects of care are strongly supported by both main protagonists of the clinical encounter can drive these stakeholders to use every opportunity available to consolidate them. Fortunately, the top priorities shared by patients and clinicians appear largely transferable to the management of other prevalent chronic physical health problems, such as diabetes or respiratory diseases. Secondly, this study sheds light on the organizational balance that primary care settings must maintain. Our results indicate that pursuing the primary care reform goals of interdisciplinary, team-based, proactive, and guideline-concordant care for chronic conditions should not occur at the expense of relational continuity, short-term responsiveness, and ensuring that healthcare professionals have the skills and flexibility to genuinely listen and tailor care to each of their patients. Thirdly, the shared priority of having a single common medical record between all healthcare providers underscores the importance of health data integration at a moment when, in Canada and elsewhere, major investments are being made in the computerization of healthcare systems often in the absence of a cohesive and standardized approach [50, 51]. Finally, the existence of divergent opinions between patients and clinicians should prompt decision-makers to ensure that both are on the same page whenever changes in care processes or services are being considered.

### Strengths and limitations

This study had several strengths. We used a structured and rigorous communication approach that prevented framing effects and important biases in group reasoning. Participation requirements were relatively minimal, and the anonymous and asynchronous process allowed a diverse group of knowledgeable participants from urban, suburban, and more remote settings to provide input into the organization of primary cardiovascular care. Our panelists proved to be very engaged and submitted many items and explanations to strengthen the group communication process. Addressing health issues from multiple perspectives is increasingly valued in the health ecosystem, but this can represent a significant challenge at a time of growing social polarization. Our study demonstrated the feasibility and appropriateness of the Delphi method for mobilizing and combining clinical and experiential knowledge in a democratic and non-threatening environment. Including faculty patient partners alongside untrained patients was another strength of our study. Experienced patient partners play an invaluable role in healthcare system renewal, but they are unlikely and should not be expected to cover the wealth and breadth of patient voices. In our study, patient partners appeared to have indirectly enabled "untrained" patients by eliciting organizational aspects that they could rally behind. Furthermore, they may have ultimately fostered the connection between patient and clinician perspectives by standing somewhere between the two, having an intimate experience of illness as well as an increased understanding of how the health system works. Finally, this was one of the first Delphi studies to our knowledge to benefit from the synergy between the rating and ranking techniques. Their combined use ensured that the items emerging as final top priorities were simultaneously of high importance and of a higher order of priority compared to the other items assessed. From a knowledge translation perspective, highlighting a smaller number of core priorities is probably a fruitful strategy for saturated primary care settings.

Despite these strengths, our results should be interpreted with several limitations in mind. First, our study was restricted to a single jurisdiction. The top organizational priorities of patients and clinicians may be different in other contexts, especially where healthcare systems differ significantly from the Canadian system in which cost for all medically necessary healthcare services is universally covered. A second limitation is related to the healthcare professionals that were involved in our study, which were mostly family physicians and nurses. Although they may be the pivotal primary care clinicians for most patients, the priorities identified in our study probably do not reflect those of allied healthcare professionals (psychologists, pharmacists, social workers), who may have favored more strongly the items related to their inclusion and collaboration with the clinical team as well as the services network in which they are embedded. For patients, participation in our study required a minimum level of digital literacy or having access to personal assistance to complete the questionnaires online. Regular internet use is ubiquitous in the adult Quebec population (at around 95%) [52]. However, this requirement may have excluded some of the most vulnerable individuals, whose perspectives on the organization of primary cardiovascular care are as important as any other, if not more. Most of our patient panelists had a postsecondary education, and our results should be interpreted with this in mind. Finally, the comparisons between patients' and clinicians' assessments in this study should be interpreted with caution due to small sample size and multiple testing [53]. Although many statistically significant differences had p-values well below the conventional threshold and all differences found were supported by other research and plausible explanations, they should be viewed as exploratory.

## Conclusions

This study identified the top six priorities shared by patients and clinicians regarding the organization of primary cardiovascular care in Quebec FMG clinics. These aspects of care—

listening and tailoring care to each patient, providing personalized information to patients, being able to rapidly reach a professional in the event of a problem, keeping professional skills up-to-date, and ensuring relational and informational continuity of care—should be kept at the forefront of primary care settings' concerns to ensure that care for cardiovascular—and possibly other chronic—health problems is delivered optimally for patients. A few areas where patients' and clinicians' perspectives differed significantly were also highlighted. Our patient panelists were less likely than clinicians to suggest aspects related to interdisciplinary team composition or the external services network as potential priorities, and they did not view offering healthy lifestyle and prevention activities at the clinic as favorably as clinicians. However, they prioritized being able to get an appointment with their family doctor on short notice and ensuring effective collaboration with specialist physicians much higher than clinicians.

## Supporting information

**S1 Table. Quantitative results for the organizational items assessed during round two (n = 33 panelists).**
(DOCX)

## Acknowledgments

We would like to extend our sincere gratitude to each Delphi panelist who agreed to participate and share their insights during this study as well as to the persons who pretested our study instruments. We would also like to thank Jocelyne Gagné for the administrative support provided during the study, Magali Girard for her helpful advice during the preparation phase of the study, as well as two anonymous reviewers for their insightful comments which allowed us to improve the manuscript.

## Author Contributions

**Conceptualization:** Claudio Del Grande, Janusz Kaczorowski, Marie-Pascale Pomey.

**Data curation:** Claudio Del Grande.

**Formal analysis:** Claudio Del Grande.

**Investigation:** Claudio Del Grande.

**Methodology:** Claudio Del Grande, Janusz Kaczorowski, Marie-Pascale Pomey.

**Project administration:** Claudio Del Grande, Janusz Kaczorowski.

**Supervision:** Janusz Kaczorowski, Marie-Pascale Pomey.

**Validation:** Claudio Del Grande, Janusz Kaczorowski, Marie-Pascale Pomey.

**Writing – original draft:** Claudio Del Grande.

**Writing – review & editing:** Claudio Del Grande, Janusz Kaczorowski, Marie-Pascale Pomey.

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
