## [Decision Letter · Decision Letter 0]

4 Nov 2022

PONE-D-22-17745What are the top priorities of patients and clinicians for the organization of primary cardiovascular care in Quebec? A modified e-Delphi studyPLOS ONE

Dear Dr. Del Grande,

Thank you for submitting your manuscript to PLOS ONE. After careful consideration, we feel that it has merit but does not fully meet PLOS ONE’s publication criteria as it currently stands. Therefore, we invite you to submit a revised version of the manuscript that addresses the points raised during the review process.

ACADEMIC EDITOR:Thank you. We are interested in accepting your submission after the minor comments below from the reviewer are addressed. 

We look forward to receiving your revised manuscript.

Kind regards,

Fares Alahdab

Academic Editor

PLOS ONE

Journal Requirements:

Reviewers' comments:

Reviewer's Responses to Questions

**Comments to the Author**

1. Is the manuscript technically sound, and do the data support the conclusions?

Reviewer #1: Yes

Reviewer #2: Yes

2. Has the statistical analysis been performed appropriately and rigorously? 

Reviewer #1: Yes

Reviewer #2: Yes

3. Have the authors made all data underlying the findings in their manuscript fully available?

Reviewer #1: Yes

Reviewer #2: Yes

4. Is the manuscript presented in an intelligible fashion and written in standard English?

Reviewer #1: Yes

Reviewer #2: Yes

5. Review Comments to the Author

Reviewer #1: The paper by Del Grande and coauthors covers an important topic and is both interesting and well written. The Delphi process that they used was appropriate to the task at hand.

One can argue whether the findings were surprising. Patients will absolutely want to be seen in a timely fashion by physicians who are personable, communicative, well-organized and up-to-date in their field, all of which I have sought when I have been a patient. The lower priority given by study patients for healthy lifestyle and prevention activities at their primary health care clinics might surprise some clinicians. But patients have other sources for such information; and I could see where, with limited time to see their health care provider, they would wish to focus on the matter at hand rather than hear (yet again) why they need to eat better and exercise more.

Nonetheless, even if at least some of the results might seem expected, the fact that there are divergences of opinion between patients and providers around organizational priorities (among other dimensions) -- ones that are not invariably self-evident -- emphasizes how important such work is to inform clinic and program planning. While the authors list as a limitation the fact that findings relating to the Quebec health care environment might incompletely translate to other settings, I rather suspect that some of the central themes will be broadly noted elsewhere.

Based on this and similar work, it seems that periodically revisiting whether patients and physicians are on the same page should be mandated, especially under circumstances where changes in process or services are being considered by health care providers. Indeed, the authors alluded to the fact that their study was conducted at the time that the COVID infection was especially rampant and disruptive. One of the side effects of the disease has been to put the practice of telemedicine into overdrive. In person clinic visits are returning, but a sizeable number of patient-clinician interactions in my cardiology subspecialty group will continue to be virtual. The work by Del Grande and colleagues shows that we should take no assumptions for granted. In my experience, televideo consultations seem generally well received, especially for routine check-ups of stable patients. But rather than make presumptions, my colleagues and I might do well to survey our patients to know how they feel and have then inform us on how we might do better.

I really do not have any significant criticisms to offer on content or writing style.

Reviewer #2: This is a well-designed and well written study, which I have read with great interest. I have some comments.

1) Can you give any reasons why six FMG clinics declined to participate? Line 131-32.

2a) Were there any changes to the questionnaires as a result of the pre-testing (before round one)?. Line 188-189.

2b) And what professions did the clinicians in the pre-test group have?

3a) Is there any rationale why the limits of 6-7 for rating and correspondingly ranking the item in the top half of its thematic list was set? Line 244-245

3b) In what way are these limits comparable?

4) Line 273, was the five nurses registered nurses or assistant nurses? And what profession had the allied health professional?

5) Among the participating patients almost all had a post-secondary education. I think readers should interpret the result with this in mind, why I suggest adding this as a limitation.

6) Add a footnote below table 1 indicating that patients encompass both patients and faculty patient partners. Line 292.

7a) Line 294-295, was the number of submitted aspects and the synthesized items equally balanced between patients and clinicians?

7b) Also, it reads: average of 4 per panelist, can this be presented for each group (patient, clinicians) also including min and max?

8) Line 361, add percentages along with the p-value 0.014

9) Line 387, the study by Wensing et al. reports patients’ involvement in decisions as one of the most important aspects, which is not so clear from your results. Or which of your identified items and themes reflect that? Please elaborate.

6. PLOS authors have the option to publish the peer review history of their article (what does this mean?). If published, this will include your full peer review and any attached files.

Reviewer #1: No

Reviewer #2: No

---

## [Author Response · Author response to Decision Letter 0]

10 Nov 2022

As stated in the updated cover letter, we appreciate the time and effort that the editor and reviewers have dedicated to providing feedback on our manuscript and are grateful for the insightful comments on and valuable improvements to our paper. We have been able to incorporate changes to reflect all of the suggestions made by the reviewers (see Response to reviewers file). We have also addressed the additional requirements requested by the editor to the best of our understanding and have uploaded the updated files.

---

## [Decision Letter · Decision Letter 1]

20 Dec 2022

What are the top priorities of patients and clinicians for the organization of primary cardiovascular care in Quebec? A modified e-Delphi study

PONE-D-22-17745R1

Dear Dr. Del Grande,

We’re pleased to inform you that your manuscript has been judged scientifically suitable for publication and will be formally accepted for publication once it meets all outstanding technical requirements.

Kind regards,

Fares Alahdab

Academic Editor

PLOS ONE

---

## [Editor Report · Acceptance letter]

26 Dec 2022

PONE-D-22-17745R1 

What are the top priorities of patients and clinicians for the organization of primary cardiovascular care in Quebec? A modified e-Delphi study 

Dear Dr. Del Grande:

I'm pleased to inform you that your manuscript has been deemed suitable for publication in PLOS ONE. Congratulations! Your manuscript is now with our production department. 

Kind regards, 

on behalf of

Dr. Fares Alahdab 

Academic Editor

PLOS ONE